# Geographic and Temporal Variability of Hepatitis E Virus Circulation in the Russian Federation

**DOI:** 10.3390/v15010037

**Published:** 2022-12-22

**Authors:** Mikhail I. Mikhailov, Anastasia A. Karlsen, Ilya A. Potemkin, Olga V. Isaeva, Vera S. Kichatova, Elena Yu. Malinnikova, Fedor A. Asadi Mobarkhan, Eugeniy V. Mullin, Maria A. Lopatukhina, Victor A. Manuylov, Elena P. Mazunina, Evgeniia N. Bykonia, Denis A. Kleymenov, Liubov I. Popova, Vladimir A. Gushchin, Artem P. Tkachuk, Andrey D. Polyakov, Ahmed Mohammed Eladly, Sergey A. Solonin, Ilya V. Gordeychuk, Karen K. Kyuregyan

**Affiliations:** 1Laboratory of Viral Hepatitis, Mechnikov Research Institute of Vaccines and Sera, 105064 Moscow, Russia; 2Department of Viral Hepatitis, Russian Medical Academy of Continuous Professional Education, 125993 Moscow, Russia; 3Medical Faculty, Belgorod State National Research University, 308015 Belgorod, Russia; 4Scientific and Educational Resource Center for High-Performance Methods of Genomic Analysis, Peoples’ Friendship University of Russia (RUDN University), 117198 Moscow, Russia; 5Gamaleya National Research Center for Epidemiology and Microbiology, 123098 Moscow, Russia; 6Skolkovo Territorial Department of Rospotrebnadzor in Moscow, 143026 Moscow, Russia; 7Botany & Microbiology Department, Faculty of Science, Al-Azhar University, Assiut 71524, Egypt; 8N.V. Sklifosovsky Research Institute for Emergency Medicine of the Moscow Health Department, 129090 Moscow, Russia; 9Chumakov Federal Scientific Center for Research and Development of Immune-and-Biological Products of Russian Academy of Sciences, 108819 Moscow, Russia

**Keywords:** hepatitis E virus, Paslahepevirus balayani, seroprevalence, molecular epidemiology, zoonosis, disease outbreaks

## Abstract

The factors influencing hepatitis E virus (HEV) circulation remain largely unexplored. We investigated HEV seroprevalence in humans and the prevalence of infection in farm pigs and rabbits in different regions of the Russian Federation, as well as the genetic diversity and population dynamics of the HEV. The anti-HEV IgG antibody detection rates in the general population increase significantly with age, from 1.5% in children and adolescents under 20 years old to 4.8% in adults aged between 20 and 59 years old to 16.7% in people aged 60 years and older. HEV seroprevalence varies between regions, with the highest rate observed in Belgorod Region (16.4% compared with the national average of 4.6%), which also has the country’s highest pig population. When compared with the archival data, both increases and declines in HEV seroprevalence have been observed within the last 10 years, depending on the study region. Virus shedding has been detected in 19 out of the 21 pig farms surveyed. On one farm, the circulation of the same viral strain for five years was documented. All the human and animal strains belonged to the HEV-3 genotype, with its clade 2 sequences being predominant in pigs. The sequences are from patients, pigs, and sewage from pig farms clustered together, suggesting a zoonotic infection in humans and possible environmental contamination. The HEV-3 population size that was predicted using SkyGrid reconstruction demonstrated exponential growth in the 1970s–1990s, with a subsequent decline followed by a short rise around the year 2010, the pattern being similar to the dynamics of the pig population in the country. The HEV-3 reproduction number (Re) that was predicted using birth–death skyline analysis has fluctuated around 1 over the past 20 years in Russia but is 10 times higher in Belgorod Region. In conclusion, the HEV-3 circulation varies both geographically and temporally, even within a single country. The possible factors contributing to this variability are largely related to the circulation of the virus among farm pigs.

## 1. Introduction

The hepatitis E virus (HEV), or the species Paslahepevirus balayani [1], is a single-stranded RNA virus that causes acute and, in some immunocompromised patients, chronic hepatitis. The epidemiology and, evidently, the pathogenicity of the HEV infection are largely dependent upon the HEV genotype [2]. Eight HEV genotypes are currently recognized [3]. Genotypes 1 and 2 are strictly anthroponotic and cause outbreaks and sporadic cases in developing countries, where poor sanitary conditions are the main factor contributing to virus circulation [4]. Other viral genotypes are able to infect different mammalian species: wild boars (genotypes 3, 5, and 6); domestic pigs (genotypes 3 and 4); deer (genotypes 3 and 4); rabbits (genotype 3ra); and camels (genotypes 7 and 8) [5]. Genotypes 3 and 4 are responsible for the autochthonous cases of HEV infection in humans in industrialized countries, with domestic pigs being recognized as a major source of infection [6]. HEV genotypes 1 to 4 are further divided into numerous sub-genotypes, with the HEV genotype 3 (HEV-3) sub-genotypes grouped into three monophyletic clades: clade 1 (3e, f, and g), clade 2 (a, b, c, h, i, j, k, l, and m), and HEV-3ra (rabbit) [3]. Interestingly, the strains from HEV-3 clade 1 were reported to be associated with more severe disease in humans compared with the HEV-3 group 2 strains [7]. Given its large territory, the Russian Federation is bordered by both non-endemic European countries where HEV-3 is tightly controlled [8], and endemic territories, such as the Central Asian countries and China, where both the zoonotic genotypes 3 and 4, as well as the anthroponotic genotype 1, are prevalent [9,10].

HEV was first isolated by the Russian virologist Mikhail Balayan in the 1980s [11], but the majority of the initial studies were devoted to HEV circulation in the southern regions of the former Soviet Union [12]. The data on current hepatitis E epidemiology in Russia are obscure. Hepatitis E has been a notifiable disease in the Russian Federation since 2013, but the annual number of reported cases varies between just 150 and 180 per year, with the vast majority of cases identified in the European part of the country, contributing to annual incidence rates of approximately 0.1 per 100,000 population [13]. These numbers are believed to be an underestimate, as the first seroprevalence studies conducted in Russia in the 1990s had already demonstrated the prevalence of antibodies to HEV (anti-HEV) in voluntary blood donors to be as high as 4% [14]. Moreover, a surge in hepatitis E incidence was recorded in one particular region in the European part of Russia, Belgorod Region, in 2011–2012, when the incidence rates were reported to be above 4.0 per 100,000 population. At that time, this exceeded the annual hepatitis A rates in the same region [15]. These data suggest that HEV may be much more prevalent in Russia than previously thought. Here, we present the results of more than 10 years of research into HEV circulation in the Russian Federation, including data on the HEV seroprevalence in humans, the prevalence of the infection in farm pigs and rabbits, and HEV genetic diversity and population dynamics.

## 2. Materials and Methods

### 2.1. Samples

All human and animal samples tested in this study are shown in Figure 1 with the indicated geographic origin. As pigs are the main reservoir of HEV in non-endemic areas, the swine herd in the studied regions is indicated in Figure 1 with a colored bar.

#### 2.1.1. Sera from Healthy Volunteers

The serum samples from 37,919 healthy volunteers from eleven regions spanning the Russian Federation from west to east were tested for anti-HEV IgG antibodies in 2018–2020. These samples were collected as a part of a large viral hepatitis serosurvey in the general population of Russia. The population sample size was calculated with the chosen power (80%) and confidence level (95%) [17] for the known size of the population of the study regions, taking into account the data on the anti-HEV antibody prevalence in neighboring countries such as Estonia, Finland, and Mongolia [18,19,20]. The subjects of the study were males and females between 0 and 95 years of age, all apparently healthy with no symptoms of acute disease at the time of enrollment in the study (either self-reported or parent-reported) and permanently resident in the study regions. Treatment using blood products within the three months before entering the study (self-reported or parent-reported) and a body temperature over 37.10 °C or acute illnesses constituted exclusion criteria. The study was made up of seven age groups, from children aged 1–14 years to senior citizens aged over 60 years (1–14, 15–19, 20–29, 30–39, 40–49, 50–59, and ≥60 years). The mean population sample size in each age group was 387 individuals (81–2751). The male/female ratio varied between 1:0.8 and 1:1.5, depending on the age group. The mean age of the participants across the entire group of volunteers was 44.2 years (SD = 22.8 years). The rural/urban population ratio varied between 1:5.5 and 1:10, depending on the region. In addition, in five of the regions surveyed (Moscow Region, Sverdlovsk Region, Tuva Republic, Sakha Republic (Yakutia), and Khabarovsk Region), serum samples from 5237 healthy volunteers were collected and tested for anti-HEV IgG antibodies in 2008, providing two time points for the HEV seroprevalence study (Figure 1). In these regions, the participants surveyed in 2008 and in 2018–2020 were not the same, but the 2008 groups included the same age groups and had similar demographics—a male/female ratio of between 1:1.1 and 1:2.4 and a rural/urban population ratio of between 1:4 and 1:10, depending on the region.

The serosurvey was conducted in accordance with the principles expressed in the Declaration of Helsinki. Written informed consent was obtained from all the participants. The study design was approved by the Ethics Committee of the Mechnikov Research Institute for Vaccines and Sera, Moscow, Russia (Approval No. 2 dated 28 February 2018) and by the Ethics Committee of the Chumakov Institute of Poliomyelitis and Viral Encephalitides, Moscow, Russia (Approval No. 91 dated 19 May 2008). All the serum samples were coded and aliquoted, and the aliquots were stored at −70 °C until testing.

#### 2.1.2. Sera from Patients with Hepatitis E

The serum samples from 22 patients diagnosed with hepatitis E (10 from Belgorod Region, 2 from Moscow, and 10 from Vladimir), obtained in 2007–2020, as well as samples from 12 hepatitis E patients involved in an outbreak in the city of Kovrov (Vladimir Region) in August 2009, were tested for anti-HEV IgM and IgG, as well as for HEV RNA. Initially, the hepatitis E in these patients was diagnosed based on anti-HEV IgM and IgG positive tests and the negative results of HAV, HBV, and HCV testing and reported to the infectious disease surveillance system. In addition, the serum samples from 41 asymptomatic inhabitants of houses where outbreak patients lived were obtained one month after the outbreak and tested for anti-HEV IgM and IgG, along with 122 archived serum samples from voluntary blood donors and pregnant women from Kovrov obtained in 2008, a year before the outbreak. All the studies on the hepatitis E patients were approved by the Ethics Committee of the Chumakov Institute of Poliomyelitis and Viral Encephalitides, Moscow, Russia (Approval No. 91 dated 19 May 2008).

#### 2.1.3. Samples from Domestic Pigs

Individual fecal samples from 2,092 pigs were collected from 21 pig farms located in seven regions of Russia between 2007 and 2016 (Figure 1). All the pigs tested were between 2 and 4 months old, except for one farm in Vladimir Region (*n* = 219), where animals aged from 0 to 12 months were surveyed to estimate age-related HEV prevalence. In one farm from Belgorod Region, samples from pigs aged between 2 and 4 months were collected repeatedly in 2012 (*n* = 74), 2013 (*n* = 75), and 2016 (*n* = 100). All the farms studied were conventional, non-closed farms purchasing gilts in combination with their own recruitment of gilts. The animals were kept together in groups of the same age and moved from department to department according to the farms’ age-sectioned systems.

#### 2.1.4. Pig Farm Sewage Samples

A total of 10 sewage samples from two pig farms were collected in Belgorod Region (five samples from one farm in 2012, and five samples from another farm in 2014) and tested for HEV RNA. The pig farms from which the sewage samples were collected were the same farms that were surveyed for HEV in the pig population. In accordance with local regulations, sewage from pig farms is stored with an added anthelmintic in large tanks isolated from the soil until it has evaporated completely, after which the solid waste is used as manure on the fields where the crops for the pigs are grown. The samples were taken from tanks with liquid, non-evaporated sewage, one sample per tank. All the sewage samples were concentrated from an initial volume of 5 L to 1 mL each, using the commercially available Virosorb-M kit (Bioservice, Moscow, Russian Federation). This method is based on the concentration of negatively charged viral particles on magnetic particles covered with polymeric silicon dioxide modified by amino groups [21]. The total nucleic acids were extracted from the concentrate with a final volume of 1 mL, using the MagNA Pure Compact Nucleic Acid Isolation Large Volume Kit I – Large Volume (Roche Applied Science, Mannheim, Germany), and subjected to HEV RNA testing as described below.

#### 2.1.5. Samples from Domestic Rabbits

Individual fecal samples from 206 farm rabbits aged between 2 and 10 months were obtained from six farms in three regions of Russia (Figure 1) in 2012–2014. The animals were kept together in groups of the same age and moved from department to department according to the farms’ age-sectioned systems.

### 2.2. Nucleic Acid Extraction

Approximately 0.5 g of fecal sample, whether from pig or rabbit, was homogenized in 10 mL of phosphate buffered saline. The homogenized samples were centrifuged at 5000× *g* for 30 min, after which the supernatants were transferred into sterile tubes and centrifuged again at 12,000× *g* for 30 min. The homogenized samples were stored at −70 °C until the nucleic acid extraction and HEV RNA testing. The RNA was isolated from the animal fecal or human serum samples of a volume of 140 µL using the QIAamp Viral RNA Mini Kit (QIAGEN, Hilden, Germany) or a volume of 200 µL using the MagNA Pure Compact Nucleic Acid Isolation Kit I (Roche Applied Science, Mannheim, Germany) and Sileks MagNA (Sileks, Moscow, Russia), following the relevant manufacturer’s protocols.

### 2.3. HEV Testing and Sequencing

The serum samples from the healthy volunteers were tested for anti-HEV IgG antibodies using commercial enzyme immunoassay (EIA) kits (DS-EIA-ANTI-HEV-G, Diagnostic Systems, Nizhniy Novgorod, Russia). All the anti-HEV IgG reactive samples were tested for anti-HEV IgM antibodies (DS-EIA-ANTI-HEV-M, Diagnostic Systems, Nizhniy Novgorod, Russia). The sera collected in 2008 were tested in the same year according to the same schedule with the same EIA kits. The testing was performed in accordance with the instructions provided by the manufacturers of the various kits used. The anti-HEV IgG test had a high specificity (94–99%) [22] and a detection limit previously shown to be 1000 mIU/mL [23].

The serum samples from the patients with hepatitis E, the fecal samples from the pigs and rabbits, and the sewage samples were tested for HEV RNA using reverse transcription polymerase chain reaction (RT-PCR), with degenerate nested primers targeting the open reading frame 2 (ORF2) region [24]. The PCR primer sequences with positions indicated for the reference strain HEV Burma (GenBank M73218) were as follows: Av1: 5′–aay tat gcm cag tac cgg gttg –3′ (outer forward, 5687–5708), Av2: 5′–ccc tta tcc tgc tga gca ttctc –3′ revers (outer reverse, 6395–6414), Av3: 5′– gty atg yty tgc ata cat ggct –3′ (inner forward 5972–5993), and Av4: 5′–agc cga cga aat yaa ttc tgt c –3′ (inner reverse 6298–6319).

All the amplified HEV fragments were excised from an agarose gel and subjected to nucleic acid isolation using a QIAquick Gel Extraction kit (QIAGEN, Hilden, Germany), in accordance with the manufacturer’s protocol. The primary nucleotide sequence was determined on the 3130 Genetic Analyzer (ABI, Foster City, CA, USA) automatic sequencer using the BigDye Terminator v3.1 Cycle Sequencing Kit, following the manufacturer’s protocol. The HEV sequences obtained in this study were deposited in the GenBank under the accession numbers HM446470, JN204462–JN204467, JX912474–JX912477 (sequences from humans), HQ380052–HQ380131, HQ399130–HQ399185, KP144127–KP144144 (sequences from pigs), and KP144111–KP144126 (sequences from rabbits).

### 2.4. Phylogenetic Analysis

The sequences obtained were aligned using MEGA 11. Phylogenetic analysis was performed for 300 nt sequences of the HEV ORF2 region (corresponds to nt positions 5996–6295, numbering by strain M73218). Analysis was performed for an HEV dataset comprising a total of 931 sequences, including all the sequences from this study, a set of reference sequences according to the International Committee on Taxonomy of Viruses (ICTV) 2022 classification, [1] a set of sub-genotype reference sequences according to the ICTV 2020 classification [3], and 562 sequences from the GenBank that were identified as HEV-3, included our region of interest, had a known year and country of sample collection, and remained after the removal of redundant samples (skip redundant with a cutoff of 5%).

Prior to conducting Bayesian phylogenetic analysis, we checked for the presence of genetic changes between the sampling time points in the HEV-3 dataset using TempEst v.1.5 software, which provided a statistically significant relationship between genetic divergence and time. A linear regression curve was observed (Appendix A), indicating a positive correlation between the genetic divergence and the sampling time, i.e., the existence of a temporal signal in the dataset that makes it sufficient to perform molecular clock analysis in order to reconstruct the evolution history of HEV-3.

#### 2.4.1. Time-Scaled Phylogenetic Analysis

Bayesian analysis was performed using the BEAST v1.10.4 software package. The Jmodeltest-2.1.10 was used to select the model. The run parameters were as follows: number of substitution schemes was 7, rate variation—+I, +G, cat 4, ML optimized. Likewise, the trial runs were performed in BEAST to select the most suitable clock model and tree prior. After all the preliminary calculations, HKY with Gamma 4 for the model, the strict clock, and the “Coalescent: Constant Size” as tree prior were selected. An initial clock rate of 9.9 × 10^−4^ subs./site/year was used for estimation purposes. During the analysis, the rate was increased to 8.3 × 10^−3^ subs./site/year. The Markov chain Monte Carlo (MCMC) method was run for 200 million generations and sampled at every 10,000 steps in two repetitions. The two parallel runs were combined using LogCombiner v1.10.4., and Tracer v1.6 was used to check for convergence. The effective sample size was >200 in both cases. The trees were annotated with TreeAnnotator v.1.10.4 using a burn-in of 10,000 trees and visualized with FigTree v.1.4.3.

#### 2.4.2. Skyline Analysis

Skyline methods were used to extract data on the HEV-3 population dynamics, namely the values of the effective number of infections and the reproduction number in Russia from the phylogenetic tree. For this purpose, the trees were built using only Russian sequences obtained in Russia and, additionally, using only sequences obtained from Belgorod Region. The analysis comprised 101 sequences collected in Russia between 2007 and 2020, including 41 sequences from Belgorod Region obtained between 2012 and 2016. Several variants of the analysis were run, but two demonstrated the best results: the Bayesian reconstruction of the celestial grid and the analysis of the horizon of birth and death.

The main parameters for both models were taken from the calculations based on the primary trees constructed using all the reference sequences. The Bayesian reconstruction of SkyGrid was performed using the BEAST v1.10.4 software to estimate the effective number of infections. For the dataset comprising all the sequences from Russia, the Bayesian coalescent SkyGrid model was used with a tree-like parameter defined as 50 and an end time point 50 years before the most recent sample. For the sequences from Belgorod Region only, parameter 20 defines the final time point 100 years before the most recent sample. The MCMC method was run for 100 million generations and sampled at every 1000 steps. Tracer v1.7.2 was used for visualization and evaluation of the quality of the run, with a burn-in of 10% (10 million generations), ESS > 1000.

The analysis of the birth and death data was carried out using the BEAST2 software to estimate the reproduction number (Re). The initial Re values were obtained from the published data [25,26]. The duration of the MCMC was set at 100 million generations. The bdskytools R-package was used to visualize the results and plotting.

### 2.5. Statistical Analysis

The data analysis was performed using graphpad.com. The statistical analysis includes an assessment of the significance of the differences in mean values between the groups using Fisher’s exact test or chi-square with Yates correction for large values (significance threshold *p* < 0.05).

## 3. Results

### 3.1. HEV Seroprevalence in the Human Population

The anti-HEV IgG antibody positivity rates in the cohorts surveyed in 2018–2020 are shown in Figure 2 and in greater detail in Appendix A. The average anti-HEV IgG antibody prevalence rate in the general population was calculated to be 4.6% (95% CI: 4.4–4.8). In several regions (Kaliningrad Region, Belgorod Region, and the Republic of Tatarstan), the anti-HEV antibody positivity rates were significantly higher than the national average, with the highest seropositivity rate (16.4% [95% CI: 14.8–18.1]) observed in Belgorod Region (Figure 2A). In all the regions, the anti-HEV antibody prevalence increased significantly with age, peaking in people over 60 years old (Figure 2B). In the vast majority of the regions studied, the significant increase in seroprevalence had a two-step pattern, rising from children and adolescents under 20 years old to adults (20–59 years) to elderly people. We therefore pooled the seroprevalence data for each region into these three age groups. On average, the anti-HEV IgG prevalence rates were 1.5% (95% CI: 1.2–1.7) in children and adolescents under 20 years old; 4.8% in adults aged between 20 and 59 years old (95% CI: 4.3–4.8); and 16.7% (95% CI: 15.4–17.9) in people aged 60 years and older. Two regions represented an exception to the general pattern: the Republic of Dagestan, where a sharp rise in seropositivity was observed only in people aged 60 years and older, and the Republic of Tatarstan, where the seropositivity rates were similarly high in the adults and in the elderly (Figure 2B). The values of the age-specific peaks varied between regions, with the highest positivity rate observed among elderly people in Belgorod Region (34.1% [95% CI: 30.0–38.4]); Kaliningrad Region (25.0% [95% CI: 17.5–34.6]); and the Republic of Dagestan (25.2% [95% CI: 21.8–29.2]). When we further stratified the group of elderly people into the subgroups 60–64, 65–69, 70–79 and ≥80 years, a significant increase in anti-HEV IgG detection rates (*p* < 0.05, Fisher’s exact test) was observed in the participants aged 80 years and older compared to the elderly people aged under 80 years (Appendix A).

A comparison of the anti-HEV IgG antibody detection rates in the groups surveyed in 2008 and from 2018 to 2020 revealed differing patterns, as shown in Figure 3. In Moscow Region, the HEV seroprevalence dropped significantly in the general population, in children and adolescents, and among elderly people (Figure 3A). A significant increase in the HEV seroprevalence was observed among the general population in Sverdlovsk Region (Figure 3B) and in Yakutia: in the latter region this increase was associated with changes in the seroprevalence within the age groups under 60 years (Figure 3D). The HEV seroprevalence remained stable in Tuva Republic and in Khabarovsk Region, both among the general population and within the individual age groups (Figure 3C,E).

To assess the proportion of those recently exposed to the virus among the seropositive individuals, all the samples which were reactive for the anti-HEV IgG antibodies were also tested for the anti-HEV IgM antibodies. On average, the proportion of individuals reactive for both the IgM and the IgG antibodies (IgM + IgG) was 0.6% (95% CI: 0.5–0.7). The proportions of the study participants reactive for both the anti-HEV IgG and the IgM antibodies in the different age groups by study region are shown in Table 1. Cases which were reactive for the anti-HEV IgM + IgG antibodies were identified in all the regions studied, with the highest positivity rate observed among the population of Belgorod Region (*p* < 0.05, chi-square with Yates correction when compared with the national average). Anti-HEV IgM + IgG reactive cases were found in almost all the age groups in all the regions, and the proportion of such cases significantly increased with age when the combined data from all the regions were analyzed (Table 1). However, a significant difference in the IgM + IgG positivity rates between the different age groups of the particular regions was only observed in a few regions. Moreover, no significant differences were observed in the proportions of IgM + IgG reactive individuals when comparing the data from the 2008 and the 2018–2021 groups (Table 1).

### 3.2. HEV Prevalence in Domestic Pigs and Rabbits

To determine the age of the piglets with a peak frequency of HEV excretion, fecal samples from 219 animals aged 0–12 months were obtained individually at a single conventional farm in the European part of the country (Vladimir Region). The samples were divided into groups corresponding to the animals’ ages: 0–4 weeks (*n* = 27); 5–8 weeks (*n* = 11); 9–12 weeks (*n* = 23); 13–16 weeks (*n* = 20); 17–20 weeks (*n* = 38); 21–26 weeks (*n* = 32); and over 27 weeks (*n* = 68). No cases of HEV excretion were observed in the animals aged between 0 and 4 weeks, nor in the animals over 20 weeks old. The peak frequency of HEV RNA detection in feces was observed in the piglets aged between 9 and 12 weeks and between 13 and 16 weeks, i.e., aged between 2 and 4 months (Table 2).

Based on these data, a subsequent survey of HEV prevalence and virus genetic diversity among farm pigs in Russia was conducted using samples from piglets aged between 2 and 4 months. Among the 21 pig farms surveyed, piglets excreting HEV were found in 19 farms from all the study regions, with the rates of HEV RNA-positive samples varying from 8.78% to 60.47%, depending on the region (Table 3). Detailed data on the HEV RNA detection rates by farm are shown in Appendix A. In Belgorod Region, one particular farm was surveyed for HEV RNA in swine feces in three separate years: 2012, 2013, and 2016. HEV RNA was detected in feces from the piglets aged between 2 and 4 months at this farm in every study year, with positivity rates of 23.0% (17 out of 74), 25.3% (19 out of 75), and 20.0% (20 out of 100) in 2012, 2013, and 2016, respectively.

In the sewage samples taken from two pig farms in Belgorod Region, HEV RNA was detected in one out of the five samples collected in 2012 at one farm, and in one out of the five samples collected in 2014 at another farm.

HEV RNA was detected in the feces of farm rabbits aged between 2 and 10 months at three farms in Moscow Region (Table 4). However, no HEV RNA-positive samples were identified at the rabbit farms surveyed in the two other regions.

The results of the analysis of the HEV sequences obtained from the animals and farm sewage are given in Section 3.4 and Section 3.5.

### 3.3. Autochthonous Cases of Hepatitis E in Humans

One-time collected serum samples from 22 patients with sporadic hepatitis E (10 from Belgorod Region, 2 from Moscow Region, and 10 from Vladimir Region) who had not traveled abroad within the six months prior to the onset of the symptoms of the disease were available for testing and were included in the study. The median age of the patients was 54 years (22 to 84 years), and the female-to-male ratio was 1:1.7. All the cases of infection were clinically pronounced, mainly with moderate disease severity (77.3% of patients); in three patients (13.6%), the disease was mild, while two patients (9.1%) developed severe hepatitis E, which in one case proved fatal. No underlying severe conditions were reported for these patients, except for one patient who had undergone a liver transplant and subsequent immunosuppressive therapy, and another patient with Burkitt’s lymphoma, both from Moscow Region and both with moderate hepatitis E.

All the sera from the patients with sporadic hepatitis E were reactive for anti-HEV IgM and IgG antibodies. HEV RNA was detected and sequenced in the sera from all the patients from Belgorod Region and Moscow Region and in 1 patient with a fatal hepatitis E outcome out of the 10 patients from Vladimir Region. All the HEV sequences belonged to HEV-3, confirming the autochthonous infection in these patients.

In addition to the sporadic disease cases, serum samples taken from 12 patients (five men and seven women) from a hepatitis E outbreak in Kovrov, a small city in Vladimir Region, were included in the study. The patients’ median age was 67 years (31 to 81). All the patients displayed mild to moderate hepatitis symptoms, including jaundice, and all were admitted to hospital over the course of three weeks in July–August 2009. All the patients were positive for anti-HEV IgM and IgG antibodies; three patients had serum HEV RNA belonging to the HEV-3 genotype. Based on the data of the epidemiological analysis, the outbreak was suspected to have been due to a contaminated water source.

### 3.4. The Molecular Epidemiology of HEV

All the human cases of autochthonous hepatitis E that tested positive for viral RNA, as well as all the cases of HEV infection in pigs in this study, were associated with genotype 3 (Figure 4). Out of the 14 HEV sequences of human origin, 12 belonged to clade 1 (sub-genotypes e, f, and g) and 2 belonged to clade 2. The vast majority of HEV sequences of swine origin also belonged to clade 1 (93.5%, 72 out of 77), and only 6.5% (5 out of 77) belonged to clade 2 (Figure 4). All the HEV sequences obtained from rabbits belonged to the HEV-ra group; no human cases were associated with this viral genotype. Generally, the swine HEV-3 sequences from the different regions formed regional clusters, although in some cases the sequences from the same region, such as Khabarovsk Region, appeared in different clusters. The sequences from the human patients and pigs, as well as the sequences from the pig farm sewage samples, grouped together, suggesting a zoonotic infection in humans (inset A in Figure 4). Interestingly, the HEV sequences from the three outbreak patients from Vladimir Region were similar but not identical to each other and were related to sequences of swine and human origin from Belgorod Region (inset A in Figure 4) but not from Vladimir Region (inset B in Figure 4). The latter can be explained by the limited number of swine HEV sequences from Vladimir Region as they were obtained from only one farm. The HEV-3 sequences obtained from one particular farm in Belgorod Region throughout 2012–2016 belonged to the same strain (inset C in Figure 4), suggesting the stable circulation of the virus in the farm settings for at least five years.

### 3.5. Reconstruction of the History of HEV Circulation and HEV Population Dynamics

We used Bayesian analysis to assess the frequency and possible directions of HEV zoonotic transmission, in order to reconstruct the history of HEV circulation in Russia and to estimate the virus population dynamics during the last few decades.

The analysis of the HEV-3 host range (Figure 5A) has shown that the main hosts for this viral genotype are humans and domestic pigs. Interestingly, many strains in clade 1, even the swine strains, have a common ancestor of human origin, while the sequences from clade 2 mainly have a common ancestor of swine origin. However, the analysis of host distribution clearly shows that the transmission of HEV-3 in two HEV-3 clades can occur in both directions, from swine to human and vice versa. Within a relatively short period of time (50 years), a particular strain can switch from the domestic pig to the human and then back again, and it can also switch to wild boar (Figure 5A). However, these results are obtained using currently available sequences, and the transmission patterns and directions may look different if sequences from additional countries are available in the future.

The phylogeographic analysis of the HEV-3 sequences shown in Figure 5B demonstrated five possible events that contributed to the HEV distribution within Russia. The first event is associated with the importation of the 3g sub-genotype around 1913 (HPD 95%: 1903–1959) from Estonia and its subsequent spread across the territory of the former Soviet Union, from Belgorod Region and Arkhangelsk Region in the west to Khabarovsk Region in the east (Figure 5B). The spread of this sub-genotype occurred mainly in domestic pigs, according to its host range, as shown in Figure 5A. Another HEV-3 variant prevalent in Russia, namely sub-genotype 3e, originated from the UK (Figure 5B) and was introduced into Russia several times, starting in 1942 (HPD 95%: 1941–1988), and then until the mid-1990s. This HEV-3 variant circulated mainly in humans (Figure 5A), at least judging by the known sequences, although it later entered the swine population and has spread among farm pigs across the country, including in Belgorod Region, where it continues to circulate steadily (Figure 5B). The third HEV-e variant prevalent in Russia, sub-genotype 3h from clade 2, is of European origin (France) and was introduced into Russia twice, in 1958 (HPD 95%: 1955–1992) and in 1987 (HPD 95%: 1983–2007). The remaining Russian strains belonging to clade 2 were detected sporadically in humans and, most likely, were associated with importations from Asian countries that did not result in subsequent circulation in Russia (Figure 5B).

Accurate phylogeographic analysis of HEV-3ra is difficult due to the limited number of HEV sequences of rabbit origin from different parts of the world. However, HEV-3ra sequences from Russia are restricted to rabbits only (Figure 5A), appear to be of Australian origin (Figure 5B), and resulted from two importations, in 2010 (HPD 95%: 2009–2013), and in 2012 (HPD 95%: 2011–2013).

Since the analysis of HEV-3 cross-species transmission showed that this is a constantly occurring bidirectional event, and the sequences of human and porcine origin represent a dynamically mixing pool, we estimated the population dynamics of HEV-3 in Russia using SkyGrid reconstruction for a single dataset, without separating the sequences depending on the host species. In addition, we performed the same analysis separately for sequences from Belgorod Region since the serosurvey data indicate that HEV circulates most intensively in this particular region. Analysis of the effective number of HEV-3 infections in Russia demonstrated exponential growth from the 1970s to the 1990s, with a subsequent decline and then a short rise around the year 2010, after which the decline continued (Figure 6A). The trend in the effective number of HEV-3 infections in Belgorod Region followed a different pattern, with a slight increase from the 1970s to the 2000s and a tendency to gradually decrease thereafter (Figure 6B).

Additionally, we calculated the reproduction number (Re) for HEV-3 in Russia and separately for HEV-3 in Belgorod Region (Figure 7), based on the population dynamics predicted using birth–death skyline analysis. The Re values indicating the number of successful infections from one infected host remained around 1 in Russia, with slight fluctuations displaying a five-year cycle pattern (Figure 7A). In contrast, the predicted Re values for HEV-3 in Belgorod Region reached almost 10 (95% HPD 3–21) and have remained constant over the past two decades (Figure 7B), suggesting a stable HEV-3 population size in this region despite the tendency to decline seen in Figure 6B.

## 4. Discussion

The seroprevalence data from this study demonstrated that HEV is prevalent in Russia, with significant regional differences in anti-HEV antibody prevalence in the general population, from 2.7% in Novosibirsk Region to 16.4% in Belgorod Region. In general, the population of the European part of the country tended to have higher anti-HEV antibody detection rates compared with the population of the Asian part. The rates and the age-specific pattern of HEV seroprevalence observed in this study are similar to those reported in other countries where HEV-3 is endemic [27,28,29]. The seroprevalence data largely depend on the sensitivity and specificity of the test used, differences in which may be the cause of the inconsistency in the reported anti-HEV antibody detection rates [30]. To avoid this possible bias, we used the same ELISA test for all the anti-HEV antibody testing.

The close relationship between increasing age and the presence of anti-HEV IgG antibodies seems to reflect cumulative exposure to the virus throughout life. This may also reflect the effect of an age group that had a higher risk of exposure to the virus several decades ago. This assumption is confirmed by the data on the possibility of the long-term preservation of post-exposure anti-HEV antibodies [31] and the serological evidence for a decrease in HEV circulation over decades [22,28,32,33]. However, the levels of naturally acquired anti-HEV antibodies and seropositivity rates in those exposed to the virus were reported to have declined over the years [34,35]. The high prevalence of anti-HEV IgG antibodies among the elderly may therefore be due to relatively recent exposure and may reflect the current circulation of HEV in older age groups. This is supported by the increase in anti-HEV IgM antibody detection rates in the elderly compared to the younger age groups observed in this study, as well as the higher numbers of symptomatic HEV infections reported among the elderly [36,37]. Moreover, in regions where the serosurvey was conducted at two time points, in 2008 and now, we observed a decline in the HEV seroprevalence rates only in Moscow Region, presumably due to the passing away of the oldest generation that could have had a higher seroprevalence rates. In other study regions, the anti-HEV antibody detection rates either remained stable or increased slightly, indicating stable HEV circulation for at least the last 10–15 years. Such trends differ significantly from the decrease in the prevalence of antibodies to the hepatitis A virus (HAV) observed recently in the same regions of the country [38], highlighting fundamental differences in the risk factors and transmission routes of HAV and HEV in Russia. On the other hand, the results of the HEV population dynamics analysis indicate a decrease in the intensity of the virus’s circulation in Russia over the past 20 years, corresponding to an estimated decline in the global population size of HEV-3 over the last 20 years [39]. Taken together, the seroprevalence data and the estimates of virus population dynamics suggest that the prevalence of HEV infection was indeed higher decades ago. However, the virus is still circulating in all age groups, contributing to morbidity in the elderly, while in some regions of Russia there are currently no signs of a decline in HEV circulation. An example of the latter is Belgorod Region, a center for pig breeding, where the country’s highest anti-HEV antibody prevalence in the general population is observed together with an estimated virus reproduction number 10 times higher than the country’s average index. The data obtained from the phylogenetic analysis of the HEV-3 sequences from this particular region suggest swine as the source of infection for humans. The results of the Bayesian analysis that took into account the host species indicate that HEV-3 cross-species transmission occurs regularly, and not only from pigs to humans, but also in the opposite direction, although the latter may be due to the lack of available ancestral sequences of porcine origin. Overall, the limited number of sequences from different hosts and different regions is an obvious limitation of this study. Another limitation is the small part of the viral genome used for analysis, although it is this genome fragment that is most widely represented in the GenBank, and its use enabled us to obtain high posterior probability values.

Although HEV-3 appeared to be highly prevalent in pigs in Russia, the relationship between the size of the pig population and the prevalence of anti-HEV antibodies in humans was not clear for the majority of the study regions, except for Belgorod Region, which is ranked first among Russian regions in terms of pig population and pork production [16]. Perhaps the reason for this is the predominance of infections in piglets that have not reached the age of slaughter, as observed in our study and in numerous studies from different countries [40]. This may be associated with the reduced risk of food-borne infection. Interestingly, the dynamics of the HEV-3 population size in Russia coincides precisely with the dynamics of the pig population in the country. The pig industry developed steadily from the 1960s to the 1990s, followed by a significant decline: in 2005, the number of pigs decreased by 2.8 times compared to that of 1990 [41]. However, measures to restore the country’s agricultural sector resulted in a 20% increase in the number of pigs, along with a 31% increase in pork production in 2008 [42]. In turn, this led to a brief increase in the HEV-3 population size around that time and possibly provoked the surge in hepatitis E incidence reported in Belgorod Region in 2011–2012 [15].

The detection of HEV RNA in wastewater from pig farms that is used to fertilize the fields around these farms suggests that not only direct contact with infected animals and consumption of undercooked pig products, but also environmental contamination, may contribute to the spread of the virus among the human populations. Evidence for the importance of the latter for HEV transmission has been described previously in other HEV-3 endemic areas [8].

Although HEV epidemics are predominantly due to the genotypes HEV-1 and HEV-2 in tropical countries, small outbreaks associated with HEV-3 or HEV-4, mainly food-borne, have been described in industrialized countries. The European Centre for Disease Prevention and Control (ECDC) reported 37 outbreaks between 2005 and 2015, with the number of cases related to outbreaks ranging from 2 to 47 cases per year [43]. Here, we have described a possible waterborne HEV outbreak associated with HEV-3. Once again, difficulties in its investigation emphasize the importance of molecular surveillance for HEV and the need for extensive sequence databases. Although waterborne outbreaks are uncommon in regions where HEV-3 is endemic, they may occur due to contamination of groundwater with HEV-bearing sewage from pig farms and the ingress of such contaminated groundwater into the water supply system if it is in a state of disrepair. Moreover, HEV-3 RNA was detected in low concentrations in tap water from properly functioning drinking water treatment plants, indicating that water may play a role in transmitting this virus [44].

The data from the current study demonstrate that the epidemiology of HEV infection in Russia exhibits the main features characteristic of the regions where HEV-3 is endemic: a high prevalence of antibodies to the virus in the population with a relatively low number of symptomatic cases; all autochthonous cases of infection are associated with HEV-3; and a high prevalence of the infection among farm pigs. The HEV-3 strains currently circulating in Russia, even those found in the east of the country in the territories adjacent to China, are of European origin and resulted from multiple introductions throughout the twentieth century, probably associated with the import of pigs from European countries. The data presented here are based on HEV (P. balayani) testing only. We did not test patient sera for the RNA of other hepeviruses, including rat HEV (Rocahepevirus ratti), which can cause infection in humans [45]. Thus, there is still a possibility that in some anti-HEV IgM positive, HEV RNA negative samples the rat HEV infection could be undiagnosed. This is a limitation of our study.

Evidently, the stable maintenance of the HEV epizootic process on pig farms, confirmed by our observation of the persistence of the same HEV-3 strain on one farm for five years and the similar data on the persistence of the virus on a pig farm in Sweden for two years [46], is the main factor ensuring a continuous source of infection with this virus among people. Therefore, to achieve control of HEV infection in countries where zoonotic HEV-3 and HEV-4 are endemic, the main goal is to break the transmission routes of the virus in pig farms and to stop its circulation there.

## 5. Conclusions

Our data demonstrated that HEV-3 circulation is inconsistent both geographically and temporally, even within one country. Due to the zoonotic nature of HEV-3 infection, the possible factors contributing to this inconsistency are largely related to the circulation of the virus among farm pigs and include: the number of pigs in a certain territory; the prevalence of HEV in farm pigs and the changes in their age-specific prevalence, especially the infection at slaughter age; and the peculiarities in pig farm practices, in particular the ways in which farm sewage is utilized. A possible human-related factor is the change in herd immunity to HEV; the decline in virus-specific immunity in adults and the elderly may contribute to an increase in the number of symptomatic cases in the future as the latter more often have symptomatic infection.

## Figures and Tables

**Figure 1 viruses-15-00037-f001:**
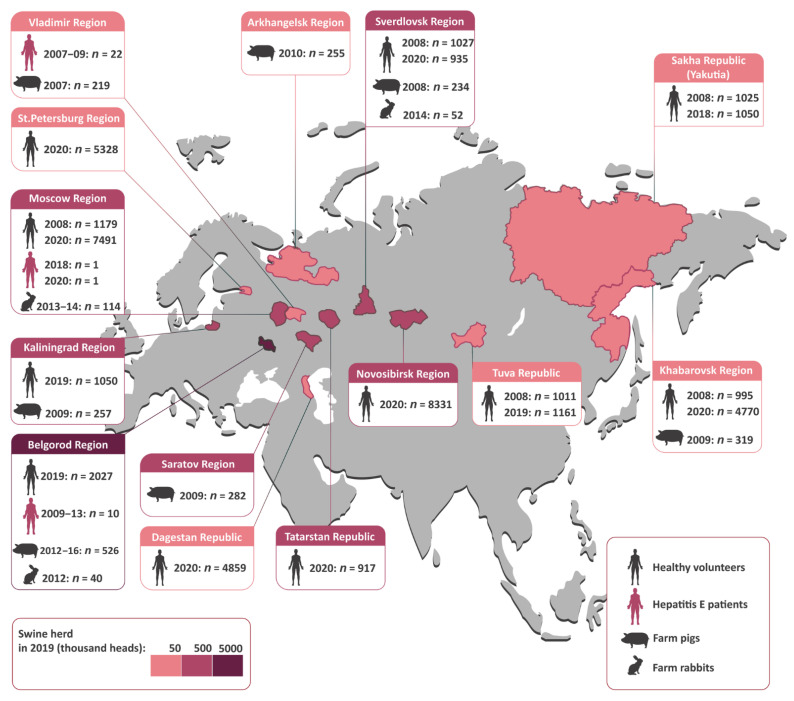
The study regions (in pink) shown on a map of Russia alongside the numbers and sources of the samples collected in each region. The colored bar represents the population of farm pigs (in thousand heads) in each study region based on a federal state statistical report from 2019 [16].

**Figure 2 viruses-15-00037-f002:**
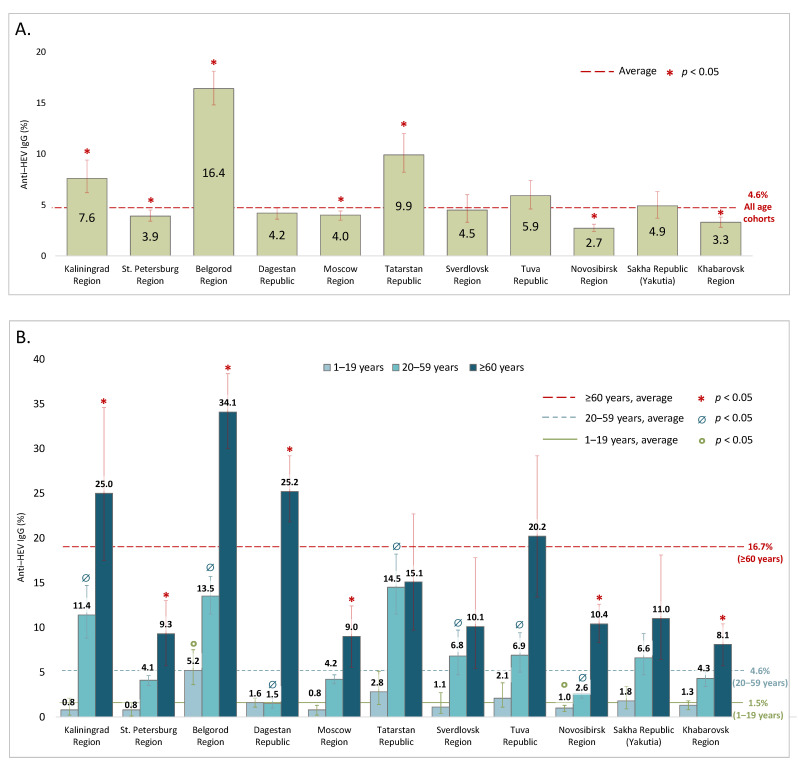
Anti-HEV IgG antibody average (**A**) and age-specific (**B**) prevalence rates among the general population in the regions studied in 2018–2020. National average and age-specific national average prevalence rates are shown with dashed lines. The statistically significant differences (chi-square with Yates correction) between regional data and national average or age-specific national average levels are shown with corresponding symbols. Error bars on the bar graphs represent 95% CI.

**Figure 3 viruses-15-00037-f003:**
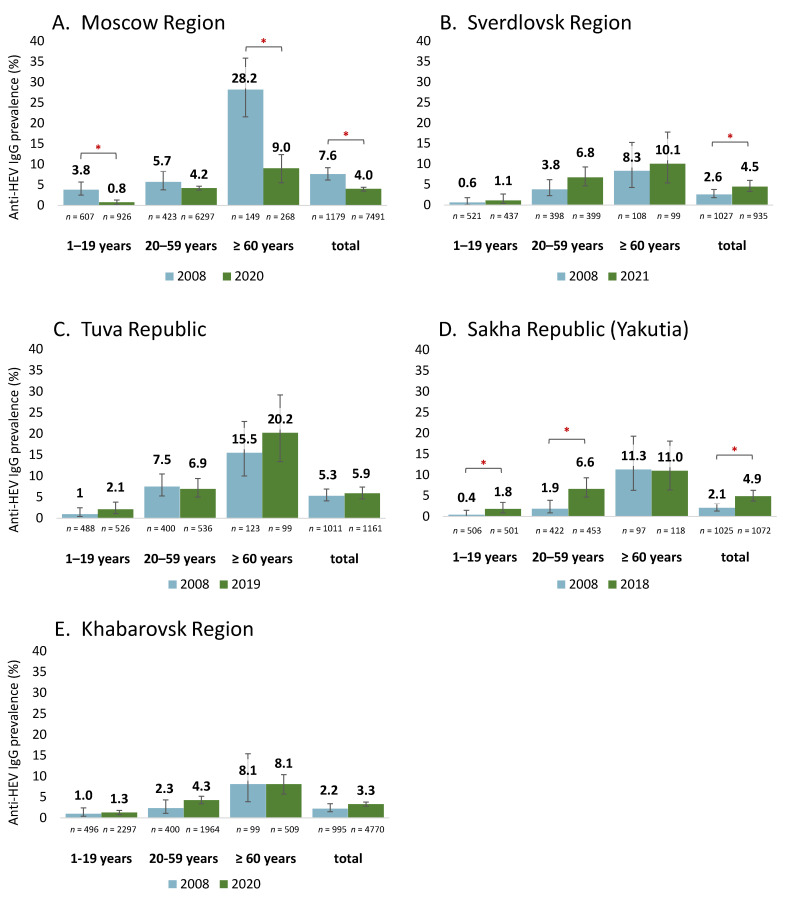
Comparison of anti-HEV IgG antibody prevalence rates in the general population of Russia in 2008 compared with 2018–2021. *p* values (Fisher’s exact test) > 0.05 between groups surveyed in different years are indicated with an asterisk. Error bars on the bar graphs represent 95% CI.

**Figure 4 viruses-15-00037-f004:**
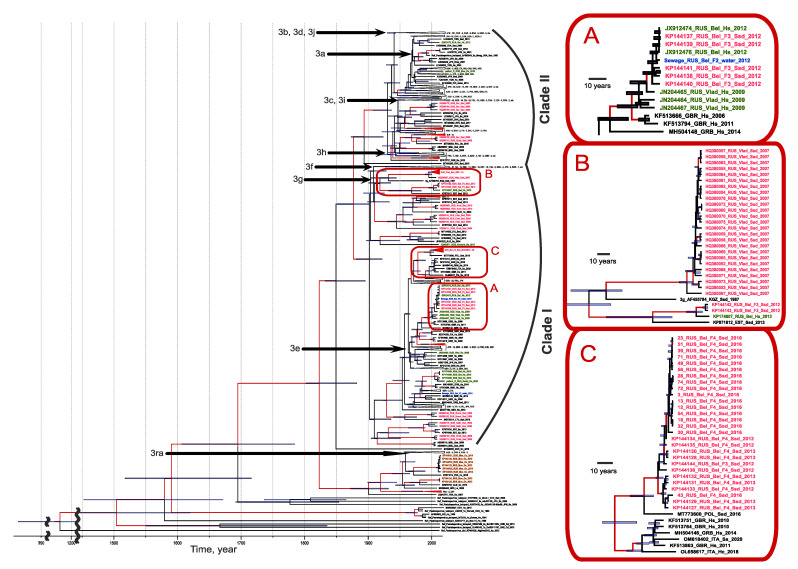
Bayesian phylogenetic tree based on 300 nt ORF-2 HEV sequences. The tree root was cut off to ensure the visibility of the modern parts of the tree. For each sequence, the number in the GenBank database, country (region), host organism, and the year of isolation are indicated. Host designations are as follows: H.s.—human (Homo sapiens), Ssd—domestic pig (Sus scrofa domesticus), Ss—wild boar (Sus scrofa), Oc—rabbit (Oryctolagus cuniculus). The sequence names from the samples collected for this study are shown in green (human), red (swine), and brown (rabbit). For compressed clusters, the number of sequences and regions of isolation are given. Tree branches with posterior probability >90% are marked in red. In each tree node, the 95% HPD is shown as a gray bar. HEV-3 sub-genotypes are indicated with arrows. The X-axis shows chronological time expressed in years. Insets (**A**–**C**) show in details clusters of interest (see explanation in the text).

**Figure 5 viruses-15-00037-f005:**
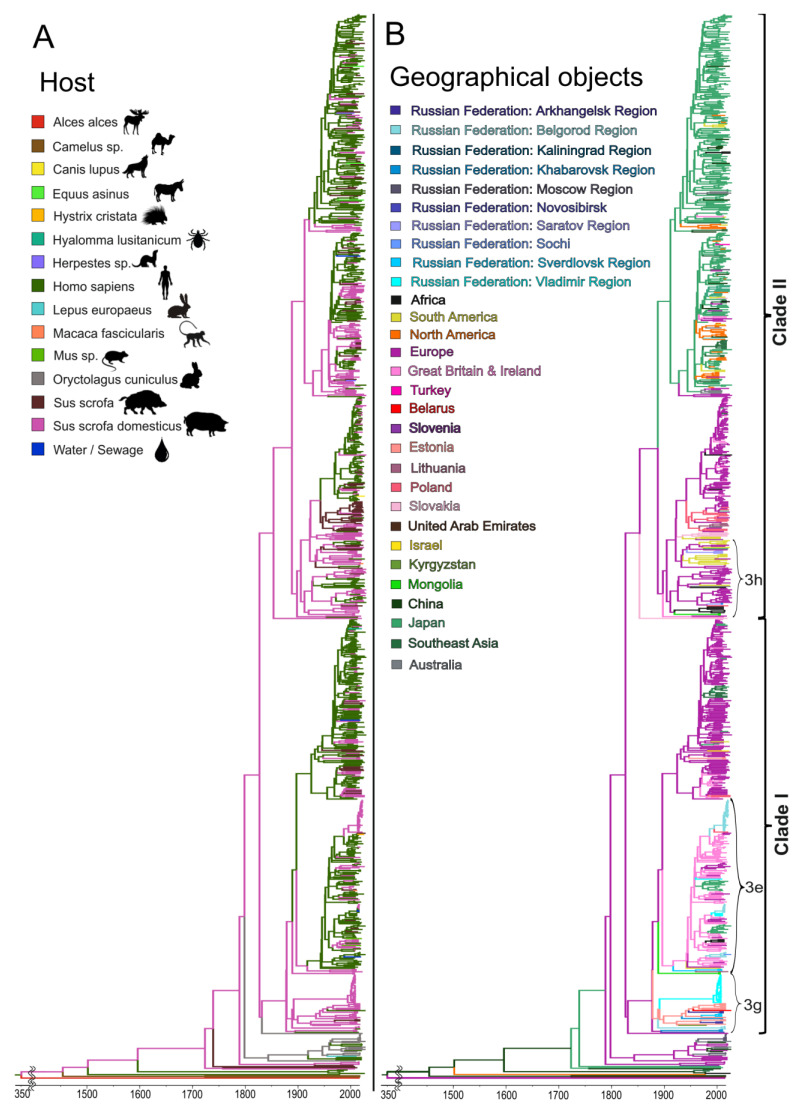
Bayesian phylogenetic tree based on 300 nt ORF-2 HEV sequences with indicated host range (**A**) and geographic region of origin (**B**). The tree root was cut off to ensure the visibility of the modern parts of the tree. The X-axis shows chronological time expressed in years.

**Figure 6 viruses-15-00037-f006:**
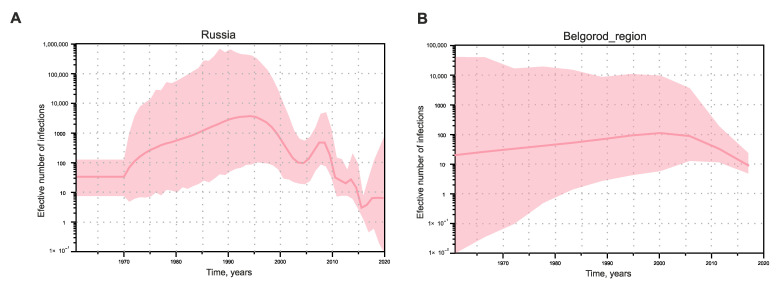
SkyGrid reconstruction for HEV-3 in Russia (**A**) and separately for Belgorod Region (**B**). The graphs show the relationship between the effective number of infections (*y*-axis) and chronological time expressed in years (*x*-axis). The red curve indicates the mean, with the 95% HPD interval shown in pink shading. Estimates were obtained using 101 sequences of the HEV-3 ORF-2 fragment (300 nt), including 41 sequences from Belgorod Region.

**Figure 7 viruses-15-00037-f007:**
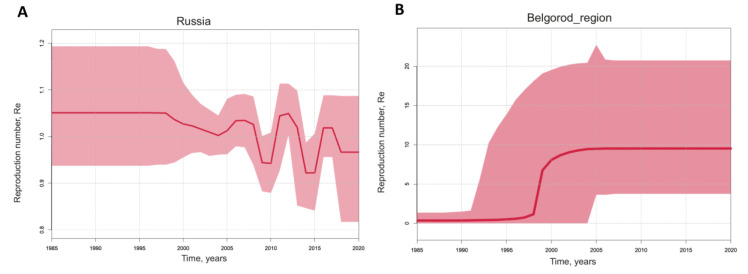
Birth-death skyline reconstruction for HEV-3 in Russia (**A**) and separately for Belgorod Region (**B**). The graphs show the relationship between the reproduction number of infection (*y*-axis) and chronological time expressed in years (*x*-axis). The curves indicate the mean, with the 95% HPD interval shown in pink shading. Estimates were obtained using 101 sequences of the HEV-3 ORF-2 fragment (300 nt), including 41 sequences from Belgorod Region.

**Table 1 viruses-15-00037-t001:** Proportion of study participants reactive for both anti-HEV IgG and IgM antibodies.

Region	Year	Anti-HEV IgG + IgM Antibodies Positive Number of Positive/Number of Tested (% [95% CI])			
		1–19 Years	20–59 Years	≥60 Years	All Age Cohorts
Kaliningrad Region	2019	0/503 (0.0 [0.0–0.9]) *	9/447 (2.0 [1.0–3.8]) ***	2/100 (2.0 [0.1–7.4])	11/1050 (1.1 [0.6–1.9])
St. Petersburg	2020	1/596 (0.2 [<0.01–1])	34/4486 (0.8 [0.5–1.1])	2/246 (0.8 [0.03–3.1])	37/5328 (0.7 [0.5–1.0])
Belgorod Region	2019	7/516 (1.4 [0.6–2.8]) ***	29/1018 (2.9 [2.0–4.1]) ***	21/493 (4.3 [2.8–6.5]) ***	57/2027 (2.8 [2.2–3.6]) ***
Republic of Dagestan	2020	2/2440 (0.1 [<0.01–0.3])	0/1889 (0.0 [0–0.2]) ***	9/530 (1.7 [0.8–3.2]) *	11/4859 (0.2 [0.1–0.4]) ***
Moscow Region	2008	2/607 (0.3 [<0.01–1.3])	3/423 (0.7 [0.1–2.2])	1/149 (0.7 [<0.01–5.0])	6/1179 (0.5 [0.2–1.1])
	2020	2/926 (0.2 [<0.01–0.8])	38/6297 (0.6 [0.4–0.8])	0/268 (0.0 [0–1.7])	40/7491 (0.5 [0.4–0.7])
Republic of Tatarstan	2020	0/364 (0.0 [0–1.3])	4/434 (0.9 [0.3–2.4])	2/119 (1.7 [0.1–6.3])	6/917 (0.7 [0.3–1.5])
Sverdlovsk Region	2008	1/521 (0.2 [<0.01–1.2])	2/398 (0.5 [0.01–1.9])	0/108 (0.0 [0–4.1])	3/1027 (0.3 [0.06–0.9])
	2021	1/437 (0.2 [0.01–1.4])	4/399 (1.0 [0.2–2.6])	0/99 (0.0 [0–4.5])	5/935 (0.5 [0.2–1.3])
Tuva Republic	2008	2/488 (0.4 [0.01–1.6])	4/400 (1.0 [0.3–2.6])	1/123 (0.8 [<0.01–5.0])	7/1011 (0.7 [0.3–1.5])
	2019	1/526 (0.2 [0.01–1.2])	1/536 (0.2 [<0.01–1.2])	3/99 (3.0 [0.7–8.9]) *	5/1161 (0.4 [0.1–1.0])
Novosibirsk Region	2020	1/3009 (0.0 [0.01–0.2])	15/4565 (0.3 [0.2–0.5]) ***	3/757 (0.4 [0.01–1.2]) ***	19/8331 (0.2 [0.1–0.4]) ***
Sakha Republic (Yakutia)	2008	0/506 (0.0 [0–0.9])	1/422 (0.2 [<0.01–1.5])	1/97 (1.0 [<0.01–6.0])	2/1025 (0.2 [<0.01–0.8])
	2018	2/501 (0.4 [0.01–1.5])	1/453 (0.2 [<0.01–1.4])	3/118 (2.5 [0.5–7.5]) *	6/1072 (0.6 [0.2–1.2])
Khabarovsk Region	2008	0/496 (0.0 [0–0.9])	1/400 (0.3 [<0.01–1.5])	1/99 (1.0 [<0.01–6.0])	2/995 (0.2 [<0.01–0.8])
	2020	9/2297 (0.4 [0.2–0.7])	16/1964 (0.8 [0.5–1.3])	2/509 (0.4 [0.01–1.5])	27/4770 (0.6 [0.4–0.8])
Average for all studied regions (with 2008 data excluded)	2018–2020	26/12,115 (0.2 [0.1–0.3]) **	151/22,488 (0.7 [0.6–0.8]) **	47/3338 (1.4 [1.0–1.8]) **	224/37,941 (0.6 [0.5–0.7])

* *p* < 0.05 (Fisher’s exact test) when compared with other age groups from the same region; ** *p* < 0.05 (chi-square with Yates correction) when compared with other age groups; *** *p* < 0.05 (chi-square with Yates correction) when compared with national average data.

**Table 2 viruses-15-00037-t002:** Frequency of HEV RNA detection in individual fecal samples from pigs of different ages at one farm in Vladimir Region.

Age of Pigs (Weeks)	Number of Tested Fecal Samples	Number of HEV RNA Positive Samples (%)
0-4	27	0 (0%)
5–8	11	2 (18.2%)
9–12	23	16 (69.6%)
13–16	20	10 (50.0%)
17–20	38	6 (15.8%)
21–26	32	0 (0%)
≥27	68	0 (0%)

**Table 3 viruses-15-00037-t003:** Rates of HEV excretion in piglets aged between 2 and 4 months.

Region	Number of Farms Surveyed	Number of Piglet fecal Samples Tested	Number of HEV RNA-Positive Samples (% [95% CI])	Number of Farms with HEV RNA-Positive Piglets
Kaliningrad Region	4	257	33 (12.84% [9.26–17.52%])	2
Arkhangelsk Region	3	255	52 (20.39% [15.88%-25.78%])	3
Belgorod Region *	4	526	115 (21.86% [18.54–25.60%])	4
Vladimir Region	1	43	26 (60.47% [45.56–73.66%])	1
Saratov Region	3	282	80 (28.37% [23.42–33.90%])	3
Sverdlovsk Region	3	234	30 (12.82% [9.09–17.75%])	3
Khabarovsk Region	3	319	28 (8.78% [6.10–12.43% ])	2

* Including data combined from surveys conducted at one farm in 2012, 2013, and 2016.

**Table 4 viruses-15-00037-t004:** Rates of HEV excretion in farm rabbits aged 2–10 months.

Region	Number of Farms Surveyed	Number of Rabbit Fecal Samples Tested	Number of HEV RNA-Positive Samples (% [95% CI])	Number of Farms with HEV RNA-Positive Rabbits
Moscow Region	3	114	9 (7.89% [4.03–14.50%])	3
Belgorod Region	2	40	0 (0.00% [0.00%-10.44%])	0
Sverdlovsk Region	1	52	0 (0.00% [0.00%-8.22%])	0

## Data Availability

The data presented in this study are available in this article and its Appendix A.

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
