# Peer review of "Geographic and Temporal Variability of Hepatitis E Virus Circulation in the Russian Federation"

_viruses, 2022, doi:10.3390/v15010037_

Round 1

Reviewer 1 Report

This manuscript by Mikhailov et al. describes important aspects of molecular epidemiology of HEV infections among humans and other animal species across 14 regions in Russia. The authors performed Bayesian analyses to estimate the time of appearance of HEV in Russia and have done a good job in describing their result in a clear and concise manner. I have no further comments for the authors. 

The article title should indicate that this study is about HEV in the Russian Federation.

Author Response

We are very grateful to Reviewer for comments and the positive opinion on our paper.

Comment: The article title should indicate that this study is about HEV in the Russian Federation

Response: We modified the article title accordingly to “Geographic and temporal variability of hepatitis E virus circulation in the Russian Federation”.

Reviewer 2 Report

This paper describes the temporal and geographical variations of HEV infections among humans, pigs, and rabbits in the Russian Federation across 14 regions. This is a molecular epidemiology study of HEV primarily in humans and pigs in Russia. Temporal variations from 2008 to 2020 among humans and pigs are described. HEV RNA from pig farms and sewage from two of these farms are described. Bayesian analyses are conducted to estimate the time of appearance of HEV in Russia, and the effective number of HEV infections and the HEV reproduction number in humans. The paper is well-written and researched.

The article title should indicate that this study is about HEV in the Russian Federation.

Figure 1. It might be useful to include thin lines connecting the inset regional boxes on this figure with the regions they represent.

Line 130. Should IgC be IgG?

Line 185. What volumes of fecal suspensions, sewage concentrates, and serum were used for RNA isolation?

Line 188. What PCR primer sequences were used for isolation of HEV RNA?

Line 218. Were any gamma rate categories used for this analysis? Were invariant sites used?

Line 242. What burn-in was used for the SkyGrid model?

The paragraph at line 432. The conclusions reached in this paragraph are contingent on the sequences used for the analysis. If sequences from additional countries had been available these conclusions may have been altered as suggested by your discussion of cross-species transmission in line 532.

Figure 6B. Although the authors concluded that there was a decrease in the effective number of infections in Belgorod after 2010, this is based on trends seen in the estimated mean effective number of infections (ENI) and the associated 95% HPD. The actual ENIs could be anywhere within the 95% HPD. While a number of lines can be drawn on this graph validating a decrease it is also possible to draw several lines across the plot that could show no decrease or even a slight rise in the ENI that would be completely within the 95% HPD. Figure 7B hints at the possibility that there is no decrease in ENI as the mean Re remains constant after 2005. However, the 95% HPD for the Re is even broader and the Re could be anywhere between 3 and 21. Because of all this the interpretation of ENI and Re for Belgorod needs to be carefully addressed.

Lines 495-457. This information would be more appropriate for the Methodology and should be moved to the ELISA section of the Methodology.

Line 558. Should HEV-2 be HEV-3 on this line?

Reference 13. Is this form on the Internet and is there a URL to this form?

Reference 26, line 693. There is no pagination/article number for this reference.

Author Response

We are very grateful to Reviewer for comments and thorough analysis of our paper. All changes made in revised manuscript are shown using changes tracking function.

Comment 1: The article title should indicate that this study is about HEV in the Russian Federation.

Response 1: We modified the article title a to “Geographic and temporal variability of hepatitis E virus circulation in the Russian Federation”.

Comment 2: Figure 1. It might be useful to include thin lines connecting the inset regional boxes on this figure with the regions they represent.

Response 2: We added lines connecting boxes with the regions on the map on this figure.

Comment 3: Line 130. Should IgC be IgG?

Response 3: Yes, it should be IgG. We fixed this typo.

Comment 4: Line 185. What volumes of fecal suspensions, sewage concentrates, and serum were used for RNA isolation?

Response 4: We added volumes used for RNA extraction to section 2.2. Nucleic acid extraction (lines 172-173 in revised manuscript). Volumes were in line with manufacturers protocols: 140 ul for QIAamp Viral RNA Mini Kit, 200 ul for MagNA Pure Compact Nucleic Acid Isolation Kit I, and 200 ul for Sileks MagNA kit. The volume used for RNA isolation from sewage concentrates was 1 ml according to manufacturer’s protocol for the kit used (MagNA Pure Compact Nucleic Acid Isolation Large Volume Kit I – Large Volume). This information is given in section 2.1.4. Pig farm sewage samples (line 158).

Comment 5: Line 188. What PCR primer sequences were used for isolation of HEV RNA?

Response 5: We added PCR primer sequences to section 2.3. HEV testing and sequencing (lines 188-192 in revised manuscript)

Comment 6: Line 218. Were any gamma rate categories used for this analysis? Were invariant sites used?

Response 6: Number of gamma categories was 4. jmodeltest-2.1.10 was used to select the model. The run parameters were as follows: Number of substitution schemes - 7, rate variation - +I, +G, cat 4, ML optimized. After all preliminary calculations, HKY with Gamma 4 for the model, the strict clock and “Coalescent: Constant Size” as tree prior were selected. We added this information Methodology section (lines 221-226 in revised manuscript)

Comment 7: Line 242. What burn-in was used for the SkyGrid model?

Response 7: Burn-in of 10% (10 million generations) was used for the SkyGrid model. We added this information Methodology section (lines 253-254 in revised manuscript)

Comment 8:

The paragraph at line 432. The conclusions reached in this paragraph are contingent on the sequences used for the analysis. If sequences from additional countries had been available these conclusions may have been altered as suggested by your discussion of cross-species transmission in line 532.

Response 8: We totally agree that sequences from additional countries could change the results of this analysis. To stress this limitation, we added this point to the paragraph (lines 438-440 in revised manuscript).

Comment 9: Figure 6B. Although the authors concluded that there was a decrease in the effective number of infections in Belgorod after 2010, this is based on trends seen in the estimated mean effective number of infections (ENI) and the associated 95% HPD. The actual ENIs could be anywhere within the 95% HPD. While a number of lines can be drawn on this graph validating a decrease it is also possible to draw several lines across the plot that could show no decrease or even a slight rise in the ENI that would be completely within the 95% HPD. Figure 7B hints at the possibility that there is no decrease in ENI as the mean Re remains constant after 2005. However, the 95% HPD for the Re is even broader and the Re could be anywhere between 3 and 21. Because of all this the interpretation of ENI and Re for Belgorod needs to be carefully addressed.

Response 9: Indeed, in the case of the Belgorod Region, the 95% HPD is broad and the actual ENI can be anywhere within the 95% HPD. Still, after the 2010, the 95% HPD become more narrow and both ENI and 95% HPD demonstrated the tendency to decrease. Re values in Belgorod, even with 95% HPD spanning from 3 to 21, are much higher compared to Russia average values. We agree, that all these data indicate the stable HEV-3 population size in Belgorod. We added this point to interpretation of results in Results section (lines 479 and 491-493 in revised manuscript).

Comment 10: Lines 495-497. This information would be more appropriate for the Methodology and should be moved to the ELISA section of the Methodology.

Response 10: We moved this information to section 2.3. HEV testing and sequencing (lines 183-184 in revised manuscript).

Comment 11: Line 558. Should HEV-2 be HEV-3 on this line?

Response 11: Yes, it should be HEV-3. We fixed this typo.

Comment 12: Reference 13. Is this form on the Internet and is there a URL to this form?

Response 12: No, this form is not available on Internet and can be obtained upon the request from Federal Center of Hygiene and Epidemiology (Russian CDC).

Comment 13: Reference 26, line 693. There is no pagination/article number for this reference.

Response 13: We added article number for this reference (line 713 in revised manuscript). No pagination is indicated in citation recommended for this reference.